# Structural Characterization of the Dimers and Selective Synthesis of the Cyclic Analogues of the Antimicrobial Peptide Cm-p5

**DOI:** 10.3390/antibiotics14020194

**Published:** 2025-02-13

**Authors:** Fidel E. Morales-Vicente, Luis A. Espinosa, Erbio Díaz-Pico, Ernesto M. Martell, Melaine Gonzalez, Gerardo Ojeda, Luis Javier González, Armando Rodríguez, Hilda E. Garay, Octavio L. Franco, Frank Rosenau, Anselmo J. Otero-González, Ludger Ständker

**Affiliations:** 1Synthetic Peptide Group, Physics and Chemistry Department, Center for Genetic Engineering and Biotechnology, P.O. Box 6162, La Habana 10600, Cuba; fidel.morales@cigb.edu.cu (F.E.M.-V.); hilda.garay@cigb.edu.cu (H.E.G.); 2Mass Spectrometry Laboratory, Systems Biology Department, Center for Genetic Engineering and Biotechnology, P.O. Box 6162, La Habana 10600, Cuba; la900415@gmail.com (L.A.E.); luis.javier@cigb.edu.cu (L.J.G.); 3Centro de Bioinformática, Simulación y Modelado (CBSM), Facultad de Ingeniería, Universidad de Talca, Talca 3460000, Chile; erbiodiaz68@gmail.com; 4Center for Protein Studies, Faculty of Biology, University of Havana, 25 Str. and I Str., La Habana 10400, Cuba; nestmartell@gmail.com (E.M.M.); glezmel93@gmail.com (M.G.); 5General Chemistry Department, Faculty of Chemistry, University of Havana, Zapata and G, La Havana 10400, Cuba; gemojedac@gmail.com; 6Core Facility for Functional Peptidomics, Ulm University Medical Center, Meyerhofstraße 4, 89081 Ulm, Germany; armando.rodriguez-alfonso@uni-ulm.de; 7Core Unit of Mass Spectrometry and Proteomics, Ulm University Medical Center, Albert-Einstein-Allee 11, 89081 Ulm, Germany; 8Centro de Analises Proteomicas e Bioquímicas, Programa de Pos-Graduaçao em Ciencias Genomicas e Biotecnologia, Universidade Catolica de Brasília, Brasília 70790-160, Brazil; ocfranco@gmail.com; 9Institute of Pharmaceutical Biotechnology, Ulm University, Albert-Einstein Alle 11, 89081 Ulm, Germany; frank.rosenau@uni-ulm.de

**Keywords:** antimicrobial peptide, SPPS, cyclization, disulfide bond, dimer, peptide sequencing

## Abstract

**Background/Objectives**: Cm-p5 and its cyclic monomeric and dimeric analogues are known for their antifungal, antibacterial, antiviral, and antibiofilm activities. Previously, our cyclization method produced a mixture of peptides that were difficult to separate, which was then improved by a selective synthesis of the parallel dimer and its differentiation from the antiparallel by comparison of the retention times in RP-HPLC. **Methods**: Here, we developed a more reliable identification method for the Cm-p5 dimer identification, which included chymotrypsin proteolytic digestion and sequencing of the different fragments by ESI-MSMS. We also improved our cyclization methods to specifically produce higher amounts of the desired cyclic variant, either cyclic monomer or dimer. **Results**: We show that liquid phase oxidation with 20% DMSO or iodine oxidation yields only the cyclic analogue. However, the on-resin oxidation with iodine showed greater efficacy and efficiency. Additionally, liquid phase cyclization yields the antiparallel dimer in high EtOH or peptide concentration, indicating a kinetic control. On the other hand, the parallel dimer was preferentially produced in 5% of TFE and low peptide concentration without the formation of the cyclic analogue indicating a thermodynamic control. **Conclusions**: In conclusion, we report that chymotryptic digestion combined with ESI-MS and MS/MS allows an unambiguous differentiation of Cm-p5 dimers. Here, we develop more selective and efficient methods for the synthesis of cyclic and dimeric analogues of Cm-p5.

## 1. Introduction

Antimicrobial peptides (AMPs) are a fundamental part of the innate immune system in all organisms [1,2]. These molecules have different activities due to distinct structural features that result in different modes of interaction with prokaryotes, fungal eukaryotes, and normal mammalian or cancer eukaryotes [3]. It is widely accepted that some of these compounds adopt amphipathic conformations that interact with the cell wall, biological membranes, alter protein folding, or inhibit enzymes, provoking metabolic destabilization and cellular stress [4]. A number of native and engineered cyclic peptides have been extensively studied and used for therapeutic applications [5]. Most natural cyclic peptides are structurally very complex. Many of them, such as nisin, can be produced by microorganisms, which reduce cost, but the vast majority must be modified to be feasible and viable for the pharmaceutical industry [6]. The engineering of natural peptides has made it possible to understand the structural basis of the mechanism of action of many of these molecules. For example, it is known that cyclization of linear peptides can improve their properties as drugs [6,7]. Nevertheless, a major drawback to the use of engineered linear AMPs compounds for medicinal purposes has been their low metabolic stability [8]. Synthetic chemists have typically overcome this limitation by using covalent modification strategies to reduce proteolytic degradation, such as *N*-alkylation, cyclization, and bioisosteric substitution [9,10]. In particular, cyclization has the advantage of reducing the conformational space of the peptide, thereby increasing the proportion of the active conformation and improving the interactions with the molecular target [11]. Many methods for peptide cyclization have been established, but the more classical macrolactamization, intermolecular disulfide bond formation, Grubbs metathesis, and azide–alkyne click reactions remain the most widely used [12,13]. Cyclization by the formation of disulfide bonds between the native cysteine residues is a powerful strategy that has added advantages of having high chemoselectivity and occurs under metal-free and mild reaction conditions [14,15]. In addition, a disulfide bond in the peptide structure influences the conformation and proteolytic stability, and enhances the pharmacological properties [16,17].

In 2015, Otero et al. reported the antifungal activity of the novel Cm-p5 peptide (SRSELIVHQRLF-NH_2_), an engineered derivative of the natural decapeptide Cm-p1, isolated from the coastal mollusk *Cenchritis muricatus* [18,19]. Recently, several amino acid substitutions and truncations were explored, and some residues relevant to the biological activity were confirmed to rationally understand the structure–activity relationships. In addition, three analogues of Cm-p5 with diverse and promising biological activities were developed, one cyclic and two dimers [20]. The cyclic version of Cm-p5 showed enhanced antifungal activity. In contrast, the antiparallel dimer showed antibacterial activity even against virulent tuberculosis and multi-resistant bacterial isolates, and the parallel dimer showed better antiviral activity [21]. In addition, all derivatives were shown to inhibit the formation of *Candida auris* biofilms at semi-inhibitory concentrations and even arresting the growth of mature biofilms [22]. Cm-p5, the cyclic, or the dimers do not show cytotoxicity against normal eukaryotic cells [20,21].

The cyclic version of Cm-p5 was prepared by replacing Glu and His residues in the sequence with Cys residues, followed by intramolecular disulfide formation under dilute conditions (0.5 mg/mL) (Appendix A). These positions were chosen because we have shown that Glu and His residues are relevant for maintaining the optimal conformation required for the antifungal activity. This folding constraint is probably mediated by a salt bridge interaction between these residues, which stabilizes the amphipathic helical structure capable of interacting with microbes [20].

Cyclization of the linear precursor CysCysCm-p5 yielded two collateral byproducts, generating a mixture of three peptides that were difficult to separate by RP-HPLC (Appendix A: zoom to Figure 18 of the SI of Reference [20]). Molecular mass measurements identified these collateral byproducts as the parallel and antiparallel dimers of CysCysCm-p5 mediated by two intermolecular disulfide bonds [20]. The selective synthesis of one of them, the parallel, using an orthogonal protection strategy allowed the differentiation of both dimers by comparing their retention times in RP-HPLC analysis. Here, we report that the combination of chymotrypsin digestion and ESI-MSMS analysis allows the unambiguous differentiation of the parallel and antiparallel dimers [23].

To further explore the therapeutic applications of these derivatives, a reliable synthetic approach must be available to obtain the cyclic peptide, as well as parallel and antiparallel dimers with high purity and yield. In addition, the optimized synthesis will facilitate novel studies of activity through the preparation of new analogues to analyze the structure–activity relationship and to characterize the mechanisms of action. This work presents the formation of the cyclic Cm-p5 monomer by using liquid oxidation with DMSO or on-resin cyclization with iodine, which is more effective and efficient. The selective synthesis of the antiparallel dimer usually requires an orthogonal protection scheme, which is more expensive [24,25]. This aspect is important to establish a feasible production process for this promising analogue. We overcame this limitation by using high concentrations of ethanol (90%) or peptide (2 mg/mL) during the air oxidation of the linear CysCysCm-p5, which allowed the exclusive formation of the antiparallel dimer. On the other hand, the parallel dimer was mainly produced at low concentrations of peptide or TFE without the formation of the cyclic monomer analogue. Future investigations will allow a deeper understanding of the chemical–physical behavior of these reactions.

## 2. Results

### 2.1. Differentiation of Cyclic Parallel and Antiparallel Dimers of Cm-p5 by Chymotryptic Digestion and ESI-MS/MS Analysis

The ESI-MS analysis of the chymotrypsin digestion of the dimer that eluted in peak 2 (Appendix A) shows two intense signals detected at *m*/*z* 394.22 (3+) and *m*/*z* 590.82 (2+) (Appendix A). These *m*/*z* values are consistent with those expected ones for the (M+3H)^3+^ and (M+2H)^2+^ protonated peptides (^1^Ser-Leu^5^)-*S*-*S*-(^6^Ile-Arg^10^) and (^1^Ser-Leu^5^)-*S*-*S*-(^6^Ile-Phe^12^-oh), which are linked by an intermolecular disulfide bond between Cys^4^-Cys^8^ and to the *C*-terminus of the original linear peptide, also deamidated by chymotrypsin. This result indicates that peak 2 contains the antiparallel dimer.

To confirm this assignment, the MS/MS spectra of the triple- and double-charged precursor ions detected at *m*/*z* 394.22 and *m*/*z* 590.82, respectively, were acquired (Figure 1a,b). The single-charged fragment ions detected at *m*/*z* 565.27 and *m*/*z* 618.34 (Figure 1a) were assigned to P1-SH and P2-SH due to the symmetric cleavage of the intermolecular disulfide bond between Cys^4^ and Cys^8^ generating the reduced peptides ^1^Ser-Leu^5^ and ^6^Ile-Arg^10^, respectively.

Several signals in Figure 1a,b were formed by the asymmetric cleavage of the disulfide bond and were assigned to 1α and 1β fragment ions (*m*/*z* 531.29 (1+) and *m*/*z* 597.25 (1+)) of the P1 peptide, and 2α and 2β fragment ions (*m*/*z* 584.35 (1+) and *m*/*z* 650.31 (1+)) of the P2 peptide (Figure 1c) with Cys^4^ and Cys^8^ residues modified as dehydroalanine and disulfohydryl cysteine [23]. In addition, further fragmentation of the backbone of 1α, 2α, and 2β fragment ions were detected at *m*/*z* 400.19 (1+), 372.20 (1+), and 438.16 (1+) and assigned to 1b4α, 2y3α, and 2y3β ions, respectively. In the P1 peptide, the mass difference of 718.30 Da between the ions detected at *m*/*z* 850.40 (1+) and 132.10 (1+) in Figure 1c and assigned as 1(2)y_2_ and 1y_1_, respectively, is consistent with the expected value (718.33 Da) for Cys^4^ in the ^1^Ser-Leu^5^ peptide, which is linked by an intermolecular disulfide bond to Cys^8^ in the ^6^Ile-Arg^10^ peptide. The same conclusion is reached when calculating the mass difference (718.33 Da) between the complementary ions at *m*/*z* 525.75 (2+) and 331.17 (1+) and assigned as 1(2)b_4_ and 1b_3_, respectively (the ion corresponding to the fragment of *m*/*z* 525.75 (2+) appears with hydroxylated *C*-terminus at 534.25 (2+)). A similar result was obtained for the P2 peptide since the mass difference of 665.28 Da between the fragment ions assigned as 2(1)y_3_ and 2y_2_ or the complementary backbone ions 2(1)b_3_ and 2b_2_ is consistent with the expected value (665.24 Da) for Cys^8^ in the ^6^Ile-Arg^10^ peptide, which is linked by an intermolecular disulfide bond to Cys^4^ in the ^1^Ser-Leu^5^peptide.

The ESI-MS analysis of the chymotrypsin digestion of the dimer eluted in peak 1 (Appendix A) showed two signals at *m*/*z* 564.29 (2+), *m*/*z* 376.51 (3+) and *m*/*z* 617.35 (2+), *m*/*z* 411.98 (3+). These *m*/*z* values were tentatively assigned to the homodimer peptides (^1^Ser-Leu^5^)-*S*-*S*-(^1^Ser-Leu^5^) and (^6^Ile-Arg^10^)-*S*-*S*-(^6^Ile-Arg^10^), which form two different disulfide bonds between Cys^4^-Cys^4^ and Cys^8^-Cys^8^, respectively (Appendix A). This result suggests the parallel dimer (Figure 2d) eluted in peak 1. To confirm this assignment, the MS/MS spectra of both homodimers were acquired (Figure 2 and Figure 3).

The MS/MS spectra of the ions detected at *m*/*z* 564.29(2+) and *m*/*z* 376.51(3+) (appearing as 359.15 (3+) due to the loss of OH) assigned to the (^1^Ser-Leu^5^)-*S*-*S*-(^1^Ser-Leu^5^) homodimer peptide are shown in Figure 2a,b. A signal detected at *m*/*z* 565.27 (1+) corresponds to the symmetric cleavage of the intermolecular disulfide bond producing the reduced ^1^Ser-Leu^5^ peptide (P1-SH or P2-SH, Figure 2c). In addition, two single-charged signals detected at *m*/*z* 531.29 and *m*/*z* 597.25 were assigned to [1α or 2α] and [1β or 2β], respectively, generated by the asymmetric cleavage of the disulfide bond containing the Cys4 modified as dehydroalanine and disulfohydryl cysteine. The mass difference of 665.28 Da between the ions detected at *m*/*z* 797.38 (1+) and 132.10 (1+) and assigned as 1(2)y_2_ or 2(1)y_2_ and 1y_1_ or 2y_1_, respectively, was in agreement with the expected value (665.26 Da) for Cys^4^ linked to the ^1^Ser-Leu^5^ peptide by an intermolecular disulfide bond. The same evidence was obtained by the analysis of b_n_ ions. All these results confirmed the presence of the homodimer (^1^Ser-Leu^5^)-*S*-*S*-(^1^Ser-Leu^5^). The remaining assignments of other fragment ions observed in the MS/MS spectra are summarized in Figure 2c.

The MS/MS spectra of the precursor ions detected at *m*/*z* 617.35 (2+) and *m*/*z* 411.89 (3+) (Figure 3a,b) also support the assignment of the (^6^Ile-Arg^10^)-*S*-*S*-(^6^Ile-Arg^10^) homodimer peptide (Figure 3c) derived from the parallel dimer of CysCysCm-p5. The signals at *m*/*z* 584.38 (1+) and 650.33 (1+) have been assigned to the asymmetric cleavage of the disulfide bond to generate the peptide (^6^Ile-Arg^10^) with the Cys^8^ converted to dehydroalanine [1α or 2α] and disulfohydryl cysteine [1β or 2β] residues, respectively. The mass difference of 718.35 Da between the 1(2)y_3_ (*m*/*z* 511.27, 2+) and 1y_2_ (*m*/*z* 303.19, 1+) fragment ions was in agreement with the expected value (718.33 Da) for Cys^8^ linked to the ^6^Ile-Arg^10^ peptide by an intermolecular disulfide bond.

### 2.2. Targeted Synthesis of the Cyclic Monomer CysCysCm-p5ss

The DMSO (20%) oxidation at pH = 3 and 1 mg/mL of CysCysCm-p5 produced only the cyclic monomer with an intramolecular disulfide bond between Cys^4^ and Cys^8^ (Figure 4). On-resin oxidation of CysCysCm-p5 with DMSO (35%) in Chemmatrix resin with 0.7 mmol/g of substitution produced the cyclic peptide only when EDT was eliminated from the cleavage mixture (Figure 4 and Figure 1). However, under these experimental conditions, the peptide yield was low (34%). The retention time changed from 19 min to 17.7 min (the original retention time was 19 min, Appendix A) due to a reduction in the length of the tubing between the column and the HPLC UV detector.

To increase the yield, the resin substitution was reduced to 0.41 mmol/g but in this case, the same cleavage cocktail without EDT (TIS 2.5%/H_2_O 2.5%) reduced the disulfide bond. We tried to reproduce the previous result by increasing the DMSO concentration up to 50% and the reaction time to 12 h, but the results were almost the same; suggesting that TIS in the cleavage cocktail reduced the intramolecular disulfide bond. In an additional experiment, where the Chemmatrix resin was replaced with a highly substituted Knorr–MBHA matrix (1.11 mmol/g), the cyclization was not possible (Appendix A).

To check the efficiency of the on-resin oxidation by DMSO, we performed a mini-cleavage with TFA 10%/TIS 1% in DCM which guarantees the preservation of the lateral chain protecting groups. We also tested the behavior of other oxidation methods, such as I_2_/DMF or O_2_/NH_3_. The RP-HPLC profiles of the cleaved peptides are shown in Appendix A and elute in a peak with higher retention times with the mass of the cyclic and protected peptide (except for the Trt protecting group of Gln).

We also evaluated two other conditions with reduced TIS content, varying water, in Chemmatrix resin, 0.41 mmol/g (Appendix A), which partially prevented the disulfide bond reduction, in the case of the lower amount of water. However, a compound that reduced the solubility of the peptide remained attached to it and changed the retention time to 23.8 min. This compound was also the cyclic peptide (Appendix A), but with a different counterion or hydrophobic adduct that modified its retention time and could not be eliminated during the column equilibration step or the purification. Direct ESI-MS measurement of the crude peptide (not collected by RP-HPLC) revealed a signal with a delta mass of 97 Da, corresponding to HSO_4−_ or H_2_PO_4−_ counterions (Appendix A).

We also tested TIS or PhSiH_3_ as one of the scavengers, but without water, in Chemmatrix resin of 0.2 mmol/g and DMSO/HCl oxidation. The results confirmed that besides a lower substitution, the elimination of water protects the cyclic product (sum of cyclic and counterion), that PhSiH_3_ is more efficient than TIS in reducing the disulfide bond under these conditions, and that it avoids the formation of adducts with counterions (Appendix A).

The result of the cleavage of the peptides in Appendix A with 94–95% TFA is shown in Appendix A (lower panels). After DMSO or O_2_ oxidation, the cleavage with 1% TIS, without water, reduced the disulfide bond by 25–30%. However, although a small number of dimers were formed, the disulfide bond formed by the oxidation of I_2_ resisted cleavage. Diastereomers are formed by Lewis acid-promoted Cys racemization, especially when the Trt protecting group is removed prior to oxidation.

Final experiments were conducted to test the use of iodine as a better oxidation method but in the inexpensive Knorr–MBHA resin. Testing the on-resin cyclization with I_2_ in MBHA 0.46 mmol/g, 0.2 g, the cyclic monomer was protected, and 100 mg of crude was obtained (62%). In 0.13 g of MBHA resin, 1.11 mmol/g, and using iodine oxidation, only 80 mg (32%) of crude was produced, indicating that an increasing substitution leads to a lower yield. For both substitutions, the amount of the dimer increased while racemization remained constant (Figure 5). In order to increase the amount of dimer formation by on-resin oxidation, we used MBHA resin with higher substitution (1.35 mmol/g) and changed the solvent during the iodine oxidation, but surprisingly the amount of dimers was reduced. Unfortunately, the yield was dramatically reduced with this resin and is not recommended for this synthesis.

### 2.3. Selective Synthesis of the Antiparallel Dimer (CysCysCm-p5)^2^ss-ss

For the synthesis of the antiparallel dimer by the one-pot formation of two disulfides with iodine and Acm-protected CysAcmCysCm-p5, the cyclic peptide was quantitatively produced (Appendix A). We observed that helix-inducing solvents increased the chemoselectivity of dimer formation depending on the experimental conditions. For example, in pure water, the parallel was favored, but the cyclic was also produced, and insoluble material remained. However, experiments with 90% TFE showed that conditions of extreme helical formation favored the antiparallel (Figure 6).

Screening different concentrations of EtOH for cyclization conditions improved the result obtained with TFE in terms of using a more environmentally friendly solvent. Indeed, 81% EtOH almost exclusively generated the antiparallel dimer at a peptide concentration of 1 mg/mL (Figure 7). On the other hand, 90% TFE did not induce the formation of the antiparallel dimer only, probably due to its higher polarity. It was also observed that the reaction time for the consumption of the starting material increased with the concentration of TFE or EtOH (Figure 8).

Finally, several experiments at 2 mg/mL of peptide showed shorter reaction times to yield mostly the antiparallel dimer in 5% EtOH or other organic solvents (Figure 8). Notably, at 1 mg/mL or more, insoluble material was present during cyclization, even in the presence of organic solvent. However, for the RP-HPLC analysis, the aliquots were solubilized by dilution to compare the chromatographic profiles and to ensure the completion of the reaction until the acyclic peptide was consumed.

### 2.4. Selective Synthesis of the Parallel Dimer (CysCysCm-p5)^2^(ss)^2^

As shown in Figure 6a and Figure 7a,b, the reaction of the linear monomer at 1 mg/mL in pure water or between 20 and 50% EtOH yields mainly the parallel dimer. The results shown in Appendix A, Figure 9a–c indicate that at 0.5 mg/mL peptide concentration, this tendency is even more pronounced, since the low peptide concentration and 5% TFE or 20–25% EtOH are the chosen conditions for the synthesis of the parallel dimer.

## 3. Discussion

### 3.1. Differentiation of Cyclic Parallel and Antiparallel Dimers of Cm-p5 by ESI-MS/MS

One aim of the present investigation was to develop an MS-based method for the unambiguous differentiation between the parallel and the antiparallel dimer. In our previous work, we used an orthogonal protection scheme for the unambiguous synthesis of the parallel dimer and the comparison of its retention time in the RP-HPLC analysis to differentiate the two dimers [20]. Identifying the dimers by their retention times alone is useful for routine quality control within a known synthesis process. However, the RP-HPLC method may not be reliable when synthesis conditions are changed, since unknown impurities generated during the new reaction may alter the retention times of the expected dimers. Therefore, we envisioned a more reliable method based on the combination of proteolytic digestion and ESI-MS/MS analysis.

Parallel and antiparallel dimers have identical elemental compositions and cannot be distinguished by ESI-MS analysis on the basis of mass accuracy. However, ESI-MS analysis of proteolysis allowed their differentiation. Proteolytic digestion of the antiparallel dimer yields a unique chymotryptic peptide with Cys4-Cys8 linked by a disulfide bond, whereas in the case of the parallel dimer, chymotrypsin generates two different homodimer chymotryptic peptides with Cys4-Cys4 and Cys8-Cys8 linked by an intermolecular disulfide bond (Appendix A). Despite the broader cleavage specificity of chymotrypsin (cleaving mainly at Trp, Tyr, Phe, and Leu), it was the protease of choice due to the presence of a single cleavage site (Leu^5^) within the cyclic sequence constrained by the two intermolecular disulfide bonds present in both dimers.

In addition to the differentiation provided by the ESI-MS analysis of the proteolytic digestions, ESI-MS/MS analysis confirmed these results. The symmetric and asymmetric cleavage of the disulfide bond, as well as the mass differences of the backbone fragment ions generated by collision-induced dissociation, allowed an unambiguous identification of the molecular masses of the disulfide-bonded peptides and the connectivity of the cysteine residues (Figure 1, Figure 2 and Figure 3) [23].

### 3.2. Selective Synthesis of the Cyclic Monomer CysCysCm-p5ss

Prior to the present study, dimers and cyclic of Cm-p5 were prepared as a mixture during liquid phase cyclization at 0.5 mg/mL, pH = 8 in water/ACN (50:50). The separation of both analogues was challenging although several RP-HPLC gradients were evaluated. The lack of selectivity in the synthesis of these molecules led us to investigate different cyclization conditions to achieve this goal.

Starting with the cyclic analog of Cm-p5 (CysCysCm-p5ss), we showed that only extreme dilution could avoid dimer formation, but such a condition is not applicable in large-scale production. Considering that the presence of chaotropic agents would favor cyclization by destroying helical secondary structures, favoring twisted conformations in which thiols are closer together, we performed the DMSO oxidation at pH = 3 and only the cyclic peptide was produced at the same concentration (even at 1 mg/mL) as with oxygen at pH = 8 (Figure 4).

It is known that the reactivity of thiol to oxidation is related to the presence of the thiolate anion at pH = 8, a kinetic factor that implies that small amounts of free sulfur catalyze the reaction in a different pathway [26]. In this sense, dimers or cyclic peptides have the same kinetic factor that allows their formation, except for the necessity that exists for aggregation to be able to form dimers or oligomers. Dimers follow a second-order kinetic law reaction with respect to peptides, and helical or extended conformation should favor them. At pH = 8, some populations of Cys are present as thiolate ions (Appendix A), reducing the net positive charge of the peptide and allowing some aggregation that, together with the increase in reactivity of the thiolate, favors dimerization because of the facial amphiphilicity of the molecule. The characteristic facial amphiphilicity of this molecule in the helical conformation determines the binding of the hydrophobic face of two molecules, while the hydrophilic face containing four positive Arg residues covers the aggregate and avoids the formation of trimmers, tetramers, and larger oligomers.

On the other hand, at pH = 3, in the presence of 20% DMSO, CysCysCm-p5 is fully protonated (two Arg residues, two Cys residues, and free N-terminus), avoiding aggregation, which, together with the chaotropic effects of DMSO favoring a twisted conformation, precludes dimerization. Intramolecular cyclization follows a first-order kinetic law with respect to peptides, and twisted conformation should favor it. The DMSO oxidation avoids aggregation and favors the formation of beta-hairpin structures that facilitate intramolecular cyclization. In summary, the linear helix CysCysCm-p5 does not cycle easily between i, i + 4 due to structural constraints and the need to have solvent-generated turns to close the thiol groups. At pH = 8, dimers are favored due to the presence of thiolates and apparently larger populations of helical secondary structures which together favor aggregation and make cyclization difficult. At pH = 3, the cyclic form is favored because the molecules could not aggregate due to an overall positive charge and DMSO destroys the helix.

Considering that DMSO oxidation requires an intermediate step of concentration/DMSO elimination when compared to air oxidation, we pay attention to a recently reported on-resin simultaneous Cys deprotection/cyclization of atosiban using I_2_/DMF in polystyrene resin [24]. On-resin peptide cyclization should be favored by three factors: (i) pseudo-dilution condition of low-substituted resins, (ii) impossibility of aggregation, or (iii) undefined secondary structure of protected peptides. On-resin supported CysCysCm-p5 have protected Arg, Ser, and Gln residues, and defined secondary structures are unlikely to be present, but random coil and twisted chain conformations could favor the approximation of the intramolecular thiol groups. In principle, the on-resin oxidation with DMSO/water or ACN-THF/water mixtures should follow a similar behavior to that of I_2_/DMF; moreover, the former involves a single step (I_2_ also removes the Trt protection of Cys). DMSO and DMF are chaotropic agents that favor disordered structures such as twisted chains, and also prevent aggregation. Chemmatrix also swells more than MBHA and keeps the peptide chains further apart, preventing dimer formation. Because I_2_ has an intense color and the possibility of Lewis acid-catalyzed Cys racemization, we prefer to start with DMSO oxidation, which is faster than O_2_ oxidation (Figure 1).

With the aim of developing a solid-phase cyclization method, the complete sequence of CysCysCm-p5 was synthesized in Chemmatrix resin (0.7 mmol/g) and the Cys was deprotected by several washes with TFA 1%/TIS 1% in DCM. The resin was then cyclized with 35% DMSO and 0.37% HCl, pH = 3 in water for 3 h (Figure 1). Hydrochloric acid maximizes the oxidation potential of DMSO because the HX/DMSO combination rapidly oxidizes the peptide coming from the usual basic media of the Fmoc/tBu protocol. The cleavage of the peptide from the resin should be conducted without EDT because, once the disulfide bond is formed, EDT reduces it. The results are shown in Figure 4 and indicate that the dimer formation is prevented under these conditions.

However, we observed a poor yield of crude peptide, a drawback of Chemmatrix resin and on-resin cyclization methods, according to our experience. In contrast to the usual 50% yield, we obtained only 38 mg (34% yield) with the cleavage mixture without EDT and 54 mg (48% yield) with EDT (111 mg, 0.15 g of resin for each experiment) (Exp. 1–2, Table 1). The lower yield in the first case is due to the loss of the cyclic peptide due to its higher solubility in ether.

In the case of cyclization in Chemmatrix resin with minor substitution, besides that the peptide was correctly oxidized (Appendix A), we could only say that the cause of the reduction is the TIS/peptide ratio, since almost the same volume of cleavage mixture is used. The application of the two-step procedure for detachment/deprotection of the Rink amide resin supports the idea that TIS is responsible for the disulfide reduction during cleavage once formed during oxidation. This reduction has already been reported for the solid phase cyclization of atosiban, where it was shown that the amount of TIS and water in the cleavage cocktail must be modulated [24].

The reduction of a disulfide bond by TIS depends on TIS and water equivalents, resin substitution, and the quantity of protecting groups and linkers that generate carbocations. These factors should be regulated peptide to peptide by calculating the available TIS equivalents. Chemmatrix resin swells so much in TFA and needs at least 15 mL/g of cleavage mixture for substitution between 0.41 and 0.7 mmol/g because the swelling is almost the same, but, evidently for the resin of 0.41 mmol/g, the excess of TIS is sufficient to scavenge carbocations and also reduce disulfide.

In the experiments with lower TIS and water in the cleavage mixture, it was not possible to protect the disulfide bond from reduction by TIS, and the formation of a counterion of +97 Da (H_3_PO_4_ or H_2_SO_4_ counterions) is not desirable. Phosphoric acid is not present in our samples and has a pKa1= 2.15, meaning it does not replace TFA counterions. In the case of sulfuric acid, it should be generated by Pbf abnormal cleavage and has pKa = −10, much stronger than TFA (pKa = 0.23), so it could replace it. A plausible explanation for this experimental fact is that at reduced H_2_O equivalents, Pbf could only be scavenged by TIS, which kinetically tends to attack the C-S bond, generating a sulfonated Arg (SO_3_ = 80 Da) that is finally converted to HSO_4_- (97 Da) after hydrolysis. The use of 1% of TIS (3.55 eq) and an excess of water (3.5%) confirms its influence in disulfide reduction via TIS probably by reaction with other cations (tBu, linker, Pbf sulfocation) (Appendix A), increasing the effective quantity of TIS available for disulfide reduction.

A final test showed that the total elimination of water from the cleavage mixture was unsuccessful in protecting the disulfide bond. It is important to note that TFA absorbs at least 1% of the moisture in the environment. The more water present, the more hydronium is formed, which is the only way TFA can dissociate as its autoprotolysis (constant of 4 × 10^−14^ [27]) is unlikely to occur. The initial result with Chemmatrix could not be reproduced even with highly substituted Knorr-MBHA (Exp. 18, Table 1). Apparently, the resin matrix is involved in the reduction of the disulfide bond by TIS; for example, polyethylene glycol should trap more water by hydrogen bridge interactions.

In the case of iodine, the cyclic product was protected from the action of TIS by residual I_2_, which completely prevented the reduction of the disulfide bond besides the formation of dimers. It has been observed that I_2_, due to its Lewis acid character, causes more racemization of Cys if the Trt of Cys has been previously eliminated. In an experiment with a resin of 0.2 mmol/g and direct I_2_ deprotection of Trt, dimers and diasteromers were minimized (Appendix A).

In addition to these good results, the development of the same iodine oxidation method in the cheaper resin MBHA is desirable. In two experiments with MBHA resin, the use of iodine was confirmed as a better oxidation method, even in 2% water/2% TIS as a cleavage mixture; in any case, increased dimers and reduced yield at 1.11 mmol/g were observed (Figure 5). Surprisingly, at 1.11 mmolg MBHA resin substitution, the quantity of dimers related to 0.46 mmol/g was the same.

At this point, the results indicate that DMSO oxidation in liquid phase and iodine oxidation in MBHA resin are the best methods for producing the cyclic analogue of Cm-p5. Reduction of the disulfide bond by TIS depends on oxidation conditions, TIS and water equivalents, the volume of the cleavage cocktail, resin type and substitution, and the quantity of protecting groups and linkers that generate carbocations, which is difficult to control unless it occurs in highly substituted Chemmatrix resins.

Table 1 summarizes all the experimental conditions evaluated. The combination of I_2_ oxidation and a middle-substituted Knorr–MBHA resin (Exp. 16, Table 1) was the best choice to ensure good yield and protect the disulfide bond during cleavage. In addition, this reaction was carried out with good purity and using a yellow solvent, such as THF (Exp. 19, Table 1). On-resin iodine oxidation is advantageous in terms of yield, operational steps, cost, and applicability to other peptides. Future research would be directed to the application of these methods to other peptides with other covalent modifications; for example, lipidated peptides have limited solubility in water and a strong tendency to aggregate. Therefore, DMSO/water oxidation should be cumbersome and the on-resin cyclization could be the preferred method.

Aiming to produce the dimers by the double oxidation of CysAcmCysCm-p5 at 5 mg/mL, we developed another method for the synthesis of the cyclic (Appendix A). In this reaction, 5 eq of iodine per Acm group was added to eliminate it and activate the sulfur atom. Subsequently, a sulfur atom from another Cys(Acm) or a free Cys nucleophilically replaces the iodine in the activated sulfur, generating the disulfide bond. Since each peptide molecule has a free Cys and a protected Cys(Acm), we expected that at a high peptide concentration (even 5 mg/mL), once the sulfur atom was activated, the free Cys of another peptide molecule replace the iodine and eventually form the antiparallel dimer by intercalation of both sequences. Unfortunately, the intramolecular attack of the free thiol to the activated sulfur in the same molecule was favored in this case, suggesting that the experimental conditions took the reaction in a different direction due to factors other than concentration. For example, the HAc solution should favor the formation of a twisted/beta-peptide structure that brings the intramolecular Cys residues together to form a disulfide bond and prevent aggregation. We also tried the reaction in THF to induce the formation of a helical structure, but the cyclic monomer was always formed. Nevertheless, this cyclization method is interesting because higher peptide concentrations (5 mg/mL) can be used compared to the process under DMSO conditions (1 mg/mL peptide); however, it requires the use of the more expensive Acm protection, iodine, and acetic acid.

### 3.3. Selective Synthesis of the Dimers

Generally, dimers or oligomers are formed due to various factors such as concentration, solvent mixture, and tendency to aggregation [28]. However, in the case of peptides, the favored secondary structure could play a determining role [29]. The concentration must be optimized from case to case, but in our experience, disulfide formation of peptides can be performed at 0.3–0.5 mg/mL without major disadvantages related to dimer formation. The tendency of CysCysCm-p5 to dimerize (at 0.5 mg/mL) was partially explained previously based on the distance between thiol groups determined by the molecular dynamic simulation of the helicoidal conformation of the acyclic peptide and the different behavior of HcyHcyCm-p5 during cyclization [20]. In principle, the results indicate that the cyclization rate is limited by the energetic barrier needed to distort the helical structure because thiol groups are farther away than the covalent distance of a sulfur–sulfur bond. This behavior could imply that the helical conformation is favored in the solvent mixtures used for air oxidation (50% ACN in water). In principle, more water favors the formation of beta, twisted, and random coil structures, whereas non-solvating or non-chaotropic organic solvents favor helicoidal structures. Beta and random coil structures should also favor aggregation because of the availability of CO-NH motifs for hydrogen bond formation. The Cm-p5 helical form should only interact facially due to its amphipathicity, conferred by the hydrophobic regions where Cys are located. The polar positive region surrounding the aggregate, while the hydrophobic phases are approached internally, may be the reason why polymerization does not occur.

If the above were true, more water or chaotropic agent in the oxidation mixture (solvent with fewer tendencies to favor helical structure, DMF, DMSO, or salts like GnCl) would favor the cyclization and more non-solvating organic solvent would favor the dimerization. For example, it is known that DMSO is a stronger solvating agent and destroys helical structures, favoring beta structures [30]. The experimental result of Figure 4 may confirm in part that hypothesis because the use of DMSO as an oxidant favors only the cyclic analogue of Cm-p5.

The results of the cyclization in water or TFE corroborate these ideas because cyclic was favored by water (strong solvating inducer solvent reduces aggregation) and besides, a parallel was also formed (apolar regions of peptide suffer some grade of hydrophobic collapse) (Figure 6). As said before, pH has a dominant role in dimer formation, only explained in terms of favoring aggregation at pH = 8. Once this pH is fixed and in the presence of helical inducer solvent (aggregation inducer solvent), dimers would compete in the reaction mixture. Apparently, more polar solvent favors the parallel; however, pure water does not produce the parallel only, an indication of kinetic control for antiparallel, favored in 90% TFE. Previous computational modeling reported demonstrates that parallel dimer is more thermodynamically stable [20].

TFE is expensive; hence, we decided to study the cyclization at different concentration of EtOH (peptide at 1 mg/mL), green solvent compared to ACN, or TFE (Figure 7). These results were more conclusive and showed that above 50% EtOH, the antiparallel is favored and below 50%, the parallel. The use of 81% EtOH almost generates the antiparallel dimer (better that with TFE 90%). The scale-up of this reaction with 15 mg of peptide but directly in 90% EtOH improves the method, besides an increase in reaction time to 15 h. The formation of the antiparallel at 2 mg/mL in 90% EtOH, besides being slower, indicates that it is favored by concentration. By that reason, at 2 mg/mL of peptide but with 5% EtOH or ACN, antiparallel is favored and no acyclic material appears after 6 h (Figure 8), water increased the rate, and the antiparallel is kinetically favored.

Even though we already have developed a synthetic method of the parallel based on the mono dimerization maintaining one Cys protected with Acm in two stages, a direct method starting from the linear peptide with two free Cys is desirable. As has been showed before, in several conditions like pH = 8 and organic solvent below 50%, the parallel is favored (Figure 8), especially at 22% EtOH. In pure water, the parallel is favored, but also the cyclic, indicating that pH allows aggregation and dimerization, but the low helical induction favors the cyclic. The chemoselectivity to the parallel in water was improved by dissolving first in DMF 5% at 1 mg/mL, because this reaction always presents insoluble material and DMF has a powerful capability of dissolving peptides. Additionally, in TFE 5%, parallel is preferred, formed in more quantity than in EtOH 5%, probably due to its major polar character. In several of these experiments (Table 2), parallel should be obtained in a modest 50–60% but the best is 25% EtOH and 0.5 mg/mL, with minimal acyclic or cyclic peptide which favor subsequent purification.

To summarize, in the selective synthesis without orthogonal protection of the dimers, at pH = 8, more water favors the parallel and the cyclic, more organic solvent and concentration favor antiparallel, but at 3 mg/mL, the parallel form is preferred. A complex reaction coordinate is governing these reactions following kinetic or thermodynamic control depending on the conditions. Due to the extension of these results, theoretical calculations and several physicochemical characterizations and explanation using DC and DLS should be part of future research.

The testing of the biological activities (antifungal, antimicrobial, antibiofilm) of all derivatives is currently ongoing.

## 4. Materials and Methods

### 4.1. Materials and Reagents

All materials were washed and dried as appropriate for peptide synthesis. Polypropylene syringes (10 mL) equipped with polypropylene filters for SPPS were used under mechanical agitation, and excess solvents and reagents were removed by vacuum filtration and collected in a suitable container.

Reagents and solvents used for peptide synthesis and purification were purchased from Merck and controlled according to their specifications. These included reagent grade Fmoc amino acids, Fmoc-Rink linker, DIEA, ascorbic acid, Ac_2_O, I_2_, and MBHA–polystirene or AM–Chemmatrix resins, and analytical grade DMF, DCM, MeOH, Et_2_O, and ACN. Peptide analysis was performed with RP-HPLC grade TFA and ACN. Commercial resins were functionalized with the Fmoc-Rink linker after acetylation to the desired substitution, if necessary. A Zorbax 300SB-C18 reversed-phase column (5 um, 4.6 × 150 mm, Agilent, Santa Clara, CA, USA) was used for peptide purity analysis. Injection water from the CIGB water treatment plant (Havana, Cuba) was used for solvent preparation and sample dilutions.

### 4.2. Peptide Characterization

#### 4.2.1. Analytical RP-HPLC and Purity Determination

Purity was determined by RP-HPLC on a Shimatzu (Shimatsu Corporation, Kyoto, Japan) or EZChrom-Elite (Knauer, Berlin, Germany) instrument (LabSolutions LC/GC 5.84 or ChromGate v3.1 chromatography software) using a C18 reversed-phase column (Zorbax, 4.6 × 150 mm) at a flow rate of 0.8 mL/min, 226 nm, and a linear gradient from 5 to 60% B in 35 min. Solvent A: 0.1% TFA in water. Solvent B: 0.05% TFA in ACN. The purity of each peak was determined by integrating the area under the chromatogram curve of the most relevant peaks. Some out-of-range or out-of-context impurities were not included in the integration.

#### 4.2.2. ESI-MS

Dimer fractions (approx. 50 μg) were collected three times by RP-HPLC and dried in SpeedVac (Savant, Instruments, Inc., New York, NY, USA) for 2 h. Intact peptide and proteolytic digestion samples were dissolved in 100 μL of 60% ACN/water solution containing 0.2% HCOOH and injected directly through metal-coated borosilicate nanocapillaries (Proxeon, Odense, Denmark). Low nitrogen pressure was used to ensure a stable spray through the capillary. The capillary was inserted into the Z-spray nanoflow electrospray ion source in positive ion mode of a hybrid quadrupole time-of-flight QTOF-2™ instrument (Waters, Milford, MA, USA). ESI-MS and MS/MS mass spectra were acquired in the *m*/*z* range of 400–2000 Th with a capillary voltage set between 900 and 1200 eV and a cone voltage of 35 eV. MS/MS spectra were obtained by collision with argon gas at the appropriate collision energy. All spectra were processed using MassLynx v4.1 program (Micromass, Wilmslow, Cheshire, UK).

#### 4.2.3. Proteolytic Digestion and Desalting of Chymotryptic Peptides

Dimer fractions (approx. 50 μg) were collected three times by RP-HPLC and dried in SpeedVac (Savant, Instruments, Inc., New York, USA) for 2 h. They were then digested for 8 h at 37 °C with 1 µg of chymotrypsin (Worthington, Lakewood, NJ, USA) dissolved in 20 µL of Tris-HCl (100 mM, pH = 8). Then, 1 µL of pure HCOOH was added to stop the reaction. Chymotryptic peptides were absorbed on ZipTip-C18 (Millipore, Burlington, MA, USA), which was previously equilibrated with 0.2% HCOOH solution. Absorbed peptides were desalted with 0.2% HCOOH solution and eluted with 4 µL of 60% ACN/0.2% HCOOH in water.

### 4.3. Peptide Synthesis

Rink–MBHA–Polystirene or Rink–AM–Chemmatrix resins were used for the Fmoc/tBu strategy of SPPS. The resin substitutions were adjusted between 0.2 and 0.7 mmol/g by capping with Ac2O prior to coupling of the Rink linker. The coupling condition was Fmoc-Aa-OH/DIC/Oxyma (4 eq each) in DMF for 30 min or after negative ninhydrin test. The Fmoc protecting group was removed with 20% piperidine in DMF (2 × 10 min). The side chains of commercial amino acids were protected with Pbf for Arg; Trt for Gln and Cys; tBu for Ser; and Acm for Cys. The resin was washed with DMF (4 × 1 min) between the different steps. At the end of the amino acid couplings, the peptide-resin was washed with DMF (4 × 1 min), MeOH (4 × 1 min), and Et2O (4 × 1 min), dried in a desiccator for 24 h, and cooled at −20 °C for 30 min before cleavage.

#### 4.3.1. Peptide Deprotection and Cleavage

A cleavage cocktail with TFA/TIS/EDT/H_2_O (94:1:2.5:2.5) was added to the cooled peptide-resin (3 mL/g for MBHA-polystyrene resin or 5 mL/g for Chemmatrix resin) in 50 mL centrifuge tubes and shaken for 2 h in a laboratory shaker. The mixture was filtered into polypropylene syringes (10 mL) fitted with polyethylene frits and added over Et_2_O at −70 °C (40 mL/g of peptide-resin), homogenized by vortexing and centrifuged. The supernatant was discarded and the procedure was repeated two more times with Et_2_O at room temperature. The product was dissolved in H_2_O/ACN (7:3), frozen at −70 °C, and freeze-dried for two days (LABCONCO, Kansas City, MO, USA).

#### 4.3.2. Two-Stage Procedure for Detachment/Deprotection of Rink Amide Resin

The peptide-resin was washed, 3 × 5 min, with 10% TFA/1% TIS in DCM. The washes were collected in a round-bottom flask and the solvent was removed under reduced pressure. The product was dissolved in H_2_O/ACN (7:3), frozen at −70 °C, and freeze-dried for two days (LABCONCO).

#### 4.3.3. Liquid-Phase Peptide Cyclization

The cyclization was carried out by dissolving the crude peptides (approx. 150 mg) in H_2_O (0.1% TFA)/DMSO (4:1) at 1 mg/mL in a round-bottom flask of 300 mL. The reaction was stirred for 6 h (or until consumption of the starting material, verified by RP-HPLC and ESI-MS). The mixture was diluted 10-fold and DMSO was removed by absortion–desorption of the peptide in the LiChroprep RP-18 matrix (25–40 μm). The 40% ACN eluate was lyophilized (LABCONCO) and purified as described before.

#### 4.3.4. On-Resin Peptide Cyclization with DMSO, O_2,_ or I_2_

On-resin DMSO cyclization was conducted by adding DMSO (35% for Chemmatrix; 50% for MBHA resin) followed by water to the peptide-resin (5 mL/g total volume). To acidulate the mixture to pH = 3, 150 μL of HCl (aq.) (35%) was added and the reaction was mechanically stirred over 3 h. Finally, the peptide-resin was washed and cleaved as described before.

On-resin oxygen cyclization was conducted by swelling the peptide-resin with THF, addition of THF/H_2_O (2:1) (5 mL/g), and NH_3_ 25% to reach pH = 8–9. The reaction was mechanically stirred overnight and finally the resin was washed and prepared for the cleavage as described before.

On-resin cyclization with I_2_ was performed by adding 3 eq of iodine dissolved in DMF to the peptide-resin (5 mL/g), allowing reacting with mechanical stirring over 30 min. Finally, the resin was washed and prepared for the cleavage as described before.

#### 4.3.5. Concentrated Iodine Oxidation of Acm-Containing Peptide

The peptide containing a free Cys and Cys-Acm was dissolved in HAc (5 mg/mL) and 5 eq of I_2_ in MeOH was added dropwise until there was a persistent brown color. Water was added, 20% of the volume of HAc used, and the mixture stirred for 1 h. Iodine was quenched by adding 1 M aq. ascorbic acid dropwise until the mixture was colorless, concentrated by evaporation under reduced pressure to approximately one third of the original volume, and analyzed by RP-HPLC.

#### 4.3.6. Low-Scale Experiments with EtOH

At 0.5 mg/mL: In a 1.5 mL Eppendorf, dissolve 0.5 mg of peptide in the necessary water with 1% TFA, sonicate during 1 min, add absolute EtOH to reach 1 mL of final volume, 10–20 μL of 25% NH_3_ or until reaching pH = 8–9. and shake overnight or until reaction was completed.

At 1 mg/mL: In a 1.5 mL Eppendorf, dissolve 1 mg of peptide in the necessary water with 1% TFA, sonicate during 1 min, add 90% natural grade EtOH to reach 1 mL at the final volume, 10–20 μL of 25% NH_3,_ or until reaching pH = 8–9, and shake overnight or until reaction was completed.

#### 4.3.7. Dimerization Scale-Up to Afford the Antiparallel Dimer

At 1 mg/mL (90% EtOH): Suspend 15 mg of crude peptide in 15 mL of Natural Ethanol (Cuba Ron S.A/Lawton Havana Distillery)/Jesús Rabí Distillery) and sonicate for 1 min. Add 150 μL of 25% NH_3_ or until pH = 8–9. Close the reaction vessel (50 mL corning) and shake overnight or until reaction was completed.

At 2 mg/mL (5% EtOH): Suspend 30 mg of crude peptide in 13.5 mL water with 1% TFA and sonicate for 1 min. Add 1.5 mL of absolute EtOH to reach 15 mL of final volume, 70 μL of 25% NH_3_ or until reaching pH = 8–9. Close the reaction vessel (50 mL corning) and shake overnight or until reaction was completed.

#### 4.3.8. Dimerization Scale-Up to Afford Parallel Dimer

At 0.5 mg/mL (25% EtOH): Suspend 7 mg of crude peptide in 11.25 mL of water with 1% TFA and sonicate during 1 min. Add 3.74 mL of absolute EtOH and 70 μL of 25% NH_3_ or until reaching pH = 9. Close the reaction vessel (50 mL corning) and shake overnight or until reaction was completed.

## 5. Conclusions

To conclude, ESI-MS and -MS/MS analysis in combination with the chymotryptic digestion allowed the unambiguous differentiation of the parallel and antiparallel dimers of the antifungal peptide Cm-p5. Improved methods for the preparation of the antifungal Cm-p5 cyclic monomer by liquid phase or on-resin oxidation have been developed. This work represents the first in-depth study of the on-resin cyclization conditions and its dependence on the type of resin, substitution, oxidation method, and cleavage mixture and the protection of the disulfide by iodine. On-resin iodine oxidation is advantageous in terms of yield, operational steps, cost, and applicability to other peptides.

Finally, this is the first time that a selective synthesis without orthogonal protection of peptide dimers is reported, indicating the importance of the solvent and pH induction of the secondary structure and aggregation. It is noteworthy that dimeric but not necessarily covalently bound helices exist in proteins, toxins, chemokines, and defensins, etc. Taking into account the new method of synthesis, these kinds of covalent dimers emerge as an interesting structural motif that should be subject to revision and generalization as a structural building block of bioorganic and medicinal chemistry.

## Data Availability

Data are contained within the article and Appendix A.

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
