# Peer review of "Structural Characterization of the Dimers and Selective Synthesis of the Cyclic Analogues of the Antimicrobial Peptide Cm-p5"

_antibiotics, 2025, doi:10.3390/antibiotics14020194_

Round 1

Reviewer 1 Report

Comments and Suggestions for Authors

Fidel E. Morales-Vicenti et.al developed a characterization and selective synthesis method for the cyclic and dimeric analogue of antimicrobial peptide Cm-P5. The author systematically optimised cyclisation methods to produce desired amount of cyclic variant.  

Major comments:

1.       AMPs are known for membrane pore formation and it can also be included as other factors which contribute to the destabilization of the prokaryotic cell

2.       It would be good to have negative control to show that the fragmentation observed is due to chymotrypsin digestion and not from spontaneous degradation/autolysis of the peptide

3.       How was the stability of the cyclic monomer peptide assessed during the RP-HPLC method? Are the solvents used for the RP- HPLC method is compatible with the peptide?

4.       It would be highly appreciated if the peak of interest and other peaks are labelled in the RP-HPLC chromatogram clearly (e.g, In Figure 4 A, B and C)

5.       How is the purity of the synthesised dimers and cyclic analogues of Cm-P5 determined and it is a critical parameter that can be included in the manuscript

6.       Does the optimised cyclisation method affect the biological functionality of the peptide?

7.       Please include materials section which includes the reagents used for the experiments/reactions  

8.       The conclusion can be made short by highlighting the key points  

Minor comments:

1.       In line number 41 of the abstract there is repetition of the words, which needs to be corrected

2.       Please ensure that abbreviations used in the manuscript are spelled out the first time they appear  

3.       Please check for typographical errors throughout the manuscript 

Author Response

Major comments:

  1. AMPs are known for membrane pore formation and they can also be included as other factors which contribute to the destabilization of the prokaryotic cell.

Lines 48-52  have been rewritten with additional information on other factors that contribute to the destabilization of the prokaryotic cell.

  1. It would be good to have negative control to show that the fragmentation observed is due to chymotrypsin digestion and not from spontaneous degradation/autolysis of the peptide.

Spontaneous degradation of the peptide is not specific to the residues of Leu and the ESI/MS spectra do not show any non-specific degradation. Spontaneous degradation occurs in the residues of Glu, Asp, (Asn and Gln if deamidated). Deamidation of the Gln of this peptide is not observed under these conditions and the corresponding fragment by degradation between Gln and Arg was not observed in any of the spectra. Chymotrypsin to peptide relation is 1ug/50ug and so chymotrypsin signals were not observed.

  1. How was the stability of the cyclic monomer peptide assessed during the RP-HPLC method? Are the solvents used for the RP- HPLC method is compatible with the peptide?

This peptide is stable in the RP-HPLC conditions used. Disulfide bridge and amide bond are stable in water with TFA 0.1% at least during the hour that take the RP-HPLC chromatogram takes and usually a few hours more. Yes, those solvents are commonly used for RP-HPLC analysis and purification of peptides.

  1. It would be highly appreciated if the peak of interest and other peaks are labelled in the RP-HPLC chromatogram clearly (e.g, In Figure 4 A, B and C).

It has been corrected. In most RP-HPLC chromatograms the peaks of interest are labelled with a text box to the side of the peak. In some other chromatograms, where the peaks are similar to other closed chromatograms that are labelled, we do not use the text box for clarity and simplicity. Some other peaks are not labelled because they are small and not of interest. We have grouped the subpanels and panels of all the figures in the new version, but they have been copied as screenshots to avoid losing resolution.

  1. How is the purity of the synthesised dimers and cyclic analogues of Cm-P5 determined and it is a critical parameter that can be included in the manuscript.

Purity is determined by calculating the peak area in RP-HPLC chromatograms at 226 nm, which is the basic method for purity determination of peptides. In the Materials and Methods, we change the title of the first heading from: Analytical RP-HPLC to Analytical RP-HPLC and Purity Determination. We have also added a final sentence: The purity of the peptides was determined by comparing the peak area of the peak of interest with the total area and are included in the Tables 1 and 2 of the manuscript.

  1. Does the optimized cyclisation method affect the biological functionality of the peptide?

We have not actually confirmed the biological activity of the peptide synthesized by the new cyclization method because we consider that is not mandatory for synthetic peptides. A peptide with the correct molecular mass and highly pure by RP-HPLC can only change its activity by changing in conformation or aggregation. For a short peptide such as Cm-p5, conformational change is unlikely; peptides are not proteins that can be denatured to an inactive conformation by denaturation. Any isomerization of an L-Aa to a D-Aa is usually detected by the robust RP-HPLC characterization method, as the stereoisomeric molecules appear in different retention times. The simple disulfide bridge of Cm-p5 does not impose a critical conformational constraint that could cause an irreversible change in conformation affecting biological activity.

In the case of aggregation, this could occur in the critical stages of precipitation in ether, lyophilization, dissolution, and change to acetate counter ion or formulation. Most of these critical steps occur after purification, so any aggregates are removed during purification and aseptic filtration. The peptide identity is accurately determined by the SPPS method, the ESI-MS spectra and the RP-HPLC chromatogram and does not change during these critical steps. These are precisely the advantages of small molecules and peptides as drugs over proteins.

These peptides are at the research and development stage and they are used directly in the in-vitro assays as trifluoacetate salt, they are not formulated or used in the in-vivo assays. After RP-HPLC purification, they do not undergo any of the critical steps mentioned above, except lyophilization. During lyophilization irreversible and specific types of aggregates of a given peptide could be formed, which could affect in the immunological properties of peptides designed for vaccines or immunogens (where the activity could be affected by aggregation), but this is not our case as these peptides have therapeutic purposes. However, nothing is absolute, there could be a special case, in fact it exists in peptoids and complex cyclic structures but this is not the case, this is a small peptide with a simple cyclization. Furthermore, the biological activity is not the subject of this publication although we will take consider it in the future.

  1. Please include materials section which includes the reagents used for the experiments/reactions.

It has already been included.

  1. The conclusion can be made short by highlighting the key points.

 It has already been shortened.

Minor comments:

  1. In line number 41 of the abstract there is repetition of the words, which needs to be corrected.

It has been corrected.

  1. Please ensure that abbreviations used in the manuscript are spelled out the first time they appear.

We ensured it was done.

  1. Please check for typographical errors throughout the manuscript 

We corrected the writing in the manuscript.

Reviewer 2 Report

Comments and Suggestions for Authors

This article summarizes the synthesis and characterization of a cyclic dimer of the antimicrobial peptide Cm-p5. ​ The study focuses on developing an improved cyclization method to produce higher amounts of the cyclic dimer, while limiting unwanted side reactions. ​ Two key strengths of the paper include: (1) developing an interesting characterization method of the Cm-p5 dimer, using chymotrypsin proteolytic degradation and LC-MS/MS sequencing of the different fragments. (2) successfully developing an improved cyclization method to produce higher amounts of the desired cyclic variant, which uses a liquid phase oxidation with 20% DMSO or iodine oxidation.

However, the paper reads like a technical protocol from inexperienced peptide chemists, rather than a research article, and there are many typographical errors throughout the manuscript that need to be addressed. Although the article appears to be very thorough, the introduction does not provide sufficient context to prepare the reader for all the data that is included. Also, biological data of the optimized synthetic peptide should also be included, with appropriate controls, to further confirm successful synthesis.  

Altogether, significant improvements are needed regarding attention to scientific writing, professional data presentation in many of the figure panels, the accuracy of technical details, and the overall presentation of the significance. Specific weaknesses are summarized in the following bullet points,  but substantial improvements are needed before this article can be truly reviewed for its scientific merits.  

  1. Limited scholarship and background throughout the abstract and/or the introduction, but there is room for improvement by strengthening the discussion of : (1) native and engineered cyclic peptides themselves, instead of just their synthesis, particularly of antimicrobial peptides. (2) prior research of cm-p5 by the authors, particularly the scientific origin (natural or artificial) and any discussion of the long-term novel applications. (3) scientific motivation for optimizing the synthesis, because marginally improving the yield is not a significant basis for publication.
  2. Scientific writing needs significant improvement. At a minimum, there are many typographical errors throughout the manuscript that need to be corrected, including spelling errors (e.g., Figure 1 caption), subject–verb agreement (e.g., line 416), etc
  3. Scientific innovation is marginal (at best), particularly because the DMSO oxidation conditions are not novel. However, there is room for improvement: is there a hypothesis for the reason that the DMSO oxidation reaction is the best? What new insights are gained from this outcome?
  4. Prior to Figure 1, the manuscript needs a better concept diagram that demonstrates the parallel and antiparallel structures and the expected differences in biological activigty
  5. Figure 1 caption has several typos, but probably all of the figure captions have typographical errors
  6. MS/MS data is good, but the LC traces are simply copy and pasted from the raw data, without proper formatting to integrate with the figure subpanels. Based on the lack of attention to detail, I am not likely to trust the accuracy of the data in this paper.
  7. Supporting information is too concise. Please introduce formatting to create a cover page, table of contents page, and separate pages for each figure.
  8. There is not a Figure 18, anywhere in the main text or supporting information, which is representative of the fact that this article needs significant attention to improving the details before it can truly be reviewed for its scientific merits.
  9. Regarding the chemmatrix resin, Lines 383–406, and elsewhere in the manuscript, this is not consistent with other reports. The stability and substantial swelling of chemmatrix resin is intended to enhance yields of difficult peptides and proteins. A proper reference should be included to support the claim on 392–393. However, the inferior results reported by the authors are believable, but this is because the authors are evaluating on-resin crosslinking methods that are eventually abandoned altogether.
  10. The mention of the technical errors, Lines 395–396, and elsewhere distract from the overall science that is being presented.

Comments on the Quality of English Language

the paper reads like a technical protocol from inexperienced peptide chemists, rather than a research article, and there are many typographical errors throughout the manuscript that need to be addressed. Although the article appears to be very thorough, the introduction does not provide sufficient context to prepare the reader for all the data that is included.

significant improvements are needed regarding attention to scientific writing, professional data presentation in many of the figure panels, the accuracy of technical details, and the overall presentation of the significance.

Supporting information is too concise. Please introduce formatting to create a cover page, table of contents page, and separate pages for each figure.

Author Response

  • This article summarizes the synthesis and characterization of a cyclic dimer of the antimicrobial peptide Cm-p5. ​ The study focuses on developing an improved cyclization method to produce higher amounts of the cyclic dimer, while limiting unwanted side reactions. ​ Two key strengths of the paper include: (1) developing an interesting characterization method of the Cm-p5 dimer, using chymotrypsin proteolytic degradation and LC-MS/MS sequencing of the different fragments. (2) successfully developing an improved cyclization method to produce higher amounts of the desired cyclic variant, which uses a liquid phase oxidation with 20% DMSO or iodine oxidation.

The remarkable result of this research was the development of a solid-phase cyclization for the synthesis of the cyclic analogue, independent of the dimers; the aim is not to obtain larger quantities, it is the selective synthesis, without interference from the dimers. The only report we know on the oxidation of peptide cysteines with oxygen in resin is for Atosiban. In that case, the variables and conditions were not studied in detail as we did in the present research. The alternative method of liquid-phase cyclization in DMSO is just an additional result of this research that we have used to explain the behavior of this molecule under different reaction conditions compared to the air oxidation method that generated the dimers. On the other hand, the second novel result of this work is the selective synthesis of the dimers through a non-orthogonal protection scheme; only changes in the solvent and concentration allow the independent synthesis of the antiparallel dimer.

  • However, the paper reads like a technical protocol from inexperienced peptide chemists, rather than a research articleand there are many typographical errors throughout the manuscript that need to be addressed.

We have improved the manuscript accordingly.

  • Although the article appears to be very thorough, the introduction does not provide sufficient context to prepare the reader for all the data that is included.

We have improved the introduction in this regard.

  • Also, biological data of the optimized synthetic peptide should also be included, with appropriate controls, to further confirm successful synthesis.

We have not actually confirmed the biological activity of the peptide synthesized by the new cyclization methods because we consider that is not mandatory for synthetic peptides. A peptide with the correct and pure molecular mass and RP-HPLC purity and time retention can only change its activity by changing in conformation or aggregation. For a short peptide such as Cm-p5, conformational change is unlikely; peptides are not proteins that can be denatured to an inactive conformation by denaturation. Any isomerization of an L-Aa to a D-Aa is usually detected by the robust RP-HPLC characterization method, as the stereoisomeric molecules appear in different retention times. The simple disulfide bridge of Cm-p5 does not impose a critical conformational constraint that could cause an irreversible change in conformation affecting biological activity.

 In the case of aggregation, this could occur in the critical stages of precipitation in ether, lyophilization, dissolution, and change to acetate counter ion or formulation. Most of these critical steps occur after purification, so any aggregates are removed during purification and aseptic filtration. The peptide identity is accurately determined by the SPPS method, the ESI-MS spectra and the RP-HPLC chromatogram and does not change during these critical steps. These are precisely the advantages of small molecules and peptides as drugs over proteins.

 These peptides are at the research and development stage and they are used directly in the in-vitro assays as trifluoacetate salt, they are not formulated or used in the in-vivo assays. After RP-HPLC purification, they do not undergo any of the critical steps mentioned above, except lyophilization. During lyophilization irreversible and specific types of aggregates of a given peptide could be formed, which could affect in the immunological properties of peptides designed for vaccines or immunogens (where the activity could be affected by aggregation), but this is not our case as these peptides have therapeutic purposes. However, nothing is absolute, there could be a special case, in fact it exists in peptoids and complex cyclic structures but this is not the case, this is a small peptide with a simple cyclization. The biological activity is not the subject of this manuscript although we will analyze it in detail in the future.

  • Altogether, significant improvements are needed regarding attention to scientific writing, professional data presentation in many of the figure panels, the accuracy of technical details, and the overall presentation of the significance. Specific weaknesses are summarized in the following bullet points, but substantial improvements are needed before this article can be truly reviewed for its scientific merits.
  1. Limited scholarship and background throughout the abstract and/or the introduction, but there is room for improvement by strengthening the discussion of :

(1) native and engineered cyclic peptides themselves, instead of just their synthesis, particularly of antimicrobial peptides.

We have improved it following your suggestion. We added some sentences and two new references related to natural and engineered peptides. However, our research is related to peptide synthesis and the final focus remained on that topic.

(2) prior research of cm-p5 by the authors, particularly the scientific origin (natural or artificial) and any discussion of the long-term novel applications.

We have added the reference to Cm-p1, the natural precursor of Cm-p5. The discussion on new long-term applications of the derivatives is included in the same paragraph.

(3) scientific motivation for optimizing the synthesis, because marginally improving the yield is not a significant basis for publication.

The scientific motivation for optimizing the synthesis of these peptides is their remarkable antifungal, antibiofilm and other activities, all of which were mentioned in the introduction. However, in terms of viability or feasibility, good activity is not enough for the pharmaceutical industry, a reliable synthesis method is essential to justify the move into non-clinical research. The above synthesis method produces a mixture of dimers and cyclic peptides, which is very difficult to separate by RP-HPLC purification, making the production process more expensive. In our opinion, the remarkable improvement in yield is an important aspect of organic synthesis in drug discovery, especially in this case, which would be impossible to achieve with the previous synthesis method.

2. Scientific writing needs significant improvement. At a minimum, there are many typographical errors throughout the manuscript that need to be corrected, including spelling errors (e.g., Figure 1 caption), subject–verb agreement (e.g., line 416), etc.

We have improved the scientific writing and have removed all the typos that we found, including the spelling mistakes.

3. Scientific innovation is marginal (at best), particularly because the DMSO oxidation conditions are not novel. However, there is room for improvement: is there a hypothesis for the reason that the DMSO oxidation reaction is the best? What new insights are gained from this outcome?

 We tried to explain that DMSO oxidation is an additional result of this research. The innovations are the on-resin oxidation with iodine and the selective dimerizations in EtOH. In lines 343-361, we explain why the DMSO oxidation reaction is better than air oxidation to obtain the cyclic independently in liquid phase. However, the best method is iodine on-resin oxidation, not the liquid-phase oxidation with DMSO. We also tried to explain that the DMSO method has several disadvantages and for this reason we have developed the resin cyclization with iodine. The new insights gained from this outcome are that iodine protect the disulfide from the reduction by TIS, dimers formation are minimal with higher substituted resins besides the reduction in yield after cleavage and iodine oxidation could be done in THF . On the other side, the chemoselective dimerization in EtOH without orthogonal protection of the Cys residues is remarkable.

4. Prior to Figure 1, the manuscript needs a better concept diagram that demonstrates the parallel and antiparallel structures and the expected differences in biological activity.

The design of the structures of the dimers could be found in previous research, correctly referenced in the introduction. However, in figures 1d, 2d and 3d we present a novel schematic structural design of the dimers. The biological activities were not included in the present work, at this stage we don’t have sufficient evidence and controls to explain differences in activity between the two dimers which will be studied in detail in future works. In the ACS Omega paper of 2019, we tried to explain the improvement of the antifungal activity of the cyclic analogue in terms of the value and direction of the hydrophobic relative moment, which is not generally considered in the literature. This new paper is focused on the synthesis of antimicrobial peptides, this is why we aim to publish in this special issue of Antibiotics, the biological activities have been correctly referenced.

5. Figure 1 caption has several typos, but probably all of the figure captions have typographical errors.

We have corrected them.

6. MS/MS data is good, but the LC traces are simply copy and pasted from the raw data, without proper formatting to integrate with the figure subpanels. Based on the lack of attention to detail, I am not likely to trust the accuracy of the data in this paper.

We prefer to simply copy and paste the raw data to improve chromatogram clarity and credibility. We have changed the format used to integrate them with the subpanels of the figures in the new version. We have grouped the subpanels and panels of all the figures in the new version, but they have been copied as screenshots to avoid losing resolution. We have not shown the integrated chromatograms to avoid cluttering the image with numbers and characters. Upon request, we can send you the LC traces in your preferred format to confirm the accuracy of our data.

7. Supporting information is too concise. Please introduce formatting to create a cover page, table of contents page, and separate pages for each figure.

We introduced some formatting and included a cover page or separate pages for each figure in the supplementary information. The index was already included.

8. There is not a Figure 18, anywhere in the main text or supporting information, which is representative of the fact that this article needs significant attention to improving the details before it can truly be reviewed for its scientific merits.

Figure 18 is from the supplementary information of our previous publication in ACS Omega. We have included this explanation to avoid copyright claims. We have tried to explain it but it was not correct. Now, we have written it correctly.

9. Regarding the chemmatrix resin, Lines 383–406, and elsewhere in the manuscript, this is not consistent with other reports. The stability and substantial swelling of chemmatrix resin is intended to enhance yields of difficult peptides and proteins. A proper reference should be included to support the claim on 392–393.

As stated in other reports, Chemmatrix resin is better for difficult peptide sequences in terms of purity gain, not in terms of yield, this is a common misunderstanding in the literature which we do not confirm in our experience (https://www.sigmaaldrich.com/DE/de/technical-documents/technical-article/protein-biology/protein-labeling-and-modification/chemmatrix-resin?srsltid=AfmBOoqG2i35gjZlJs1lmf5uAZqyo1oKYBsAbQ2xAHW5PaX2dzg4NzJB). The fundamental factor to consider in peptide synthesis is purity, because there may be an excellent yield of a given peptide, however, with a poor purity. Yield is not enough, purity is mandatory. Our research focuses first on purity and also on yield. Chemmatrix resin is better for challenging sequences because of its better swelling, not precisely because of its higher stability since polystyrene resin is also stable. The peptide Cm-p5 does not have a complicated sequence, so purity is not the issue in its case, whether polystyrene or Chemmatrix resin is used. In this case, yield becomes the critical variable. Our experience with Chemmatrix allows us to confirm and emphasize the statement that Chemmatrix gives low yield compared to MBHA-Polystirene, even though we are using it in a resin-based cyclization method. To avoid misunderstandings, we improved the sentence in lines 391-393: However, we observed a low yield of the crude peptide, a drawback of Chemmatrix resin-based and resin-based cyclization methods, in our experience and reported by https://americanpeptidesociety.org/award/albericio-fernando-2/ in https:// doi.org/10.3390/mps5050072.

The statement on lines 392-393, as far as we know, has not been described by anyone, these are our experimental results, and we cannot reference it.

  • However, the inferior results reported by the authors are believable, but this is because the authors are evaluating on-resin crosslinking methods that are eventually abandoned altogether.

10. The mention of the technical errors, Lines 395–396, and elsewhere distract from the overall science that is being presented.

The sentence has been removed.

Reviewer 3 Report

Comments and Suggestions for Authors

MS is well written  and presentation is well defined. All the findings are supported with suitable analytical data.

Synthesis novelty is lacking in the MS and however the biology part of the molecule is having vast importance and hence the MS sounds good at it.

The reason behind kinetic and thermodynamic control during the parallel and antiparallel dimers need to be encountered in the MS.

line 41-demonstrate-duplicated

line 108-highly concentrated EtOH/peptide. kindly indicate the concentrations. 

MSMS spectrograms are too small to identify the mass data, kindly provide the magnified spectrums

Fig1, c and d; Fig2, c and d; Fig 3 c <COOH and NH2 are in small caps, change to capitals)

Fig 1 D ; Fig 2 D <cannot identify the peptide sequence>

Fig 4, Fig 5, 6, 7,8,9 <push the magnified data to supplementary info>

Check the overall MS for grammattical errors

Author Response

MS is well written and presentation is well defined. All the findings are supported with suitable analytical data. Synthesis novelty is lacking in the MS and however the biology part of the molecule is having vast importance and hence the MS sounds good at it.The reason behind kinetic and thermodynamic control during the parallel and antiparallel dimers need to be encountered in the MS.

  • In this article we present the results and explain them briefly due to the length that take to explain the kinetic and thermodynamic results. A more detailed explanation will be addressed in future studies. A simple explanation is already provoded in the MS: the higher the concentration, the more antiparallel, which indicates kinetic control, and therefore the antiparallel must be under thermodynamic control. It is also explained that in reference 20 a computational study was done showing that the parallel is more stable, so it is formed under thermodynamic control.

line 41-demonstrate-duplicated.

  • It has been corrected.

line 108-highly concentrated EtOH/peptide, kindly indicate the concentrations. 

  • It has been indicated.

MS/MS spectrograms are too small to identify the mass data, kindly provide the magnified spectrums.

  • We have improved it.

Fig1, c and d; Fig2, c and d; Fig 3 c -COOH and -NH2 are in small caps, change to capitals).

  • It has been corrected.

Fig 1 D; Fig 2 D <cannot identify the peptide sequence>

  • It was corrected.

Fig 4, Fig 5, 6, 7,8,9 <push the magnified data to supplementary info>

  • It has been corrected, we provide the magnified chromatograms. We have grouped the subpanels and panels of all the figures in the new version, but they have been copied as screenshots to avoid losing resolution.

Check the overall MS for grammatical errors.

  • The manuscript has been improved. 

Round 2

Reviewer 2 Report

Comments and Suggestions for Authors

Thanks for making the requested changes. They look great. 

Lines 103-105 & 114. Do you mean "therapeutic applications"? I think the phrasing of these statements should be well chosen, because they are the key justification for this paper and subsequent papers. Also, an unoptimized synthesis is not always a bottleneck because brute force can be applied. Presumably, the optimized synthesis facilitates novel studies of activity, though the  preparation of new homologs for establishing structure–activity relationships, characterizing mechanisms, and installing probes, etc.  Presumable, the new chemistry provides access to these homologs for future syntheses and studies

Based on the above comments--I would like to recommend two experiments that would finalize the rigor of the paper and provide a concrete conclusion, but I think only one needs to be included in the current paper. The combination of both experiments could be the 'next paper'

1. Including one confirmatory biological experiment, or simply an 'activity' assay of the resulting peptides. This experiment would show that the new synthesis gives functional material as expected. I assume this experiment has been done already and could easily be included.

2. Synthesis of mutated peptide sequences (i.e., substrate table), showing the generalizability of the dimerization reaction. A glycine scan or something would be sufficient. The results would be informative to you and other groups, even if none of the new dimer reactions are successful.  

Author Response

Dear Reviewer 2:

We have improved lines 103-105 & 114 according to your suggestions. Yes, we mean therapeutic applications. We have improved the manuscript accordingly and added the sentence (105-107):

“The optimized synthesis will facilitates novel studies of activity through the  preparation of new homologs to analyze the structure-activity relationship and characterize the mechanisms of action.”

Regarding the synthesis of mutated peptide sequences (substrate table; glycine scan), this is an excellent advise. We have synthesized similar analogues that confirm the kinetic and thermodynamic preference of dimer formation, but data are still preliminary. Also, the syntheses of further theoretically improved analogues is ongoing at the moment.

Related to the confirmatory biological experiments we haven't done it yet. Of course, these experiments are indispensable and we will be carried out with the required scientific quality in the future. The activity tests will include antifungal, antibacterial and antibiofilm action of the different derivatives. We have to do it in cooperation with our international collaboration partners outside of Cuba because of our limited resources, and this will take us several months. We have added a corresponding sentence in the discussion (563-564).